# Blue and Yellow Light Induce Changes in Biochemical Composition and Ultrastructure of *Limnospira fusiformis* (Cyanoprokaryota)

**DOI:** 10.3390/microorganisms11051236

**Published:** 2023-05-08

**Authors:** Matilde Pelagatti, Giovanna Mori, Sara Falsini, Raffaello Ballini, Luigi Lazzara, Alessio Papini

**Affiliations:** Department of Biology, University of Florence, Via P.A. Micheli, 1-3, 50121 Firenze, Italy; matilde.pelagatti@stud.unifi.it (M.P.); giovanna.mori@unifi.it (G.M.); raffaelloballini@gmail.com (R.B.); luigi.lazzara@unifi.it (L.L.)

**Keywords:** *Limnospira fusiformis*, *Arthrospira*, cyanobacteria, microalgal growth, phycocyanin, proteins

## Abstract

*Limnospira fusiformis* (also known as Spirulina) is a cyanobacterium that is widely cultivated due to its economic importance. It has specific pigments such as phycocyanin that allow it to grow at different light wavelengths compared to other cultivated algae. Our study investigated the effect of yellow (590 nm) and blue (460 nm) light fields on various biochemical features, including the pigment concentration, protein content, dry weight, and cell ultrastructure of *L. fusiformis*. Our findings revealed that biomass growth was faster in yellow light compared to blue light, with a higher relative amount of proteins even after one day of exposure. However, after eight days, the relative protein content in yellow versus blue light was not statistically different. Furthermore, in yellow light, we observed a decrease in chlorophyll *a*, an increase in cyanophycin granules, and an increase in the amount of dilated thylakoids. On the other hand, in blue light, there was an increase in phycocyanin after one day, along with an increase in electron-dense bodies, which are attributable to carboxysomes. However, after eight days, the differences in pigment content compared to the control were not statistically significant. Our study showed that using specific wavelengths during the harvesting phase of spirulina growth can enhance phycocyanin content with blue light (after one day) and biomass, growth rates, and protein content with yellow light after six days. This highlights the biotechnological potential of this approach.

## 1. Introduction

*Limnospira fusiformis (Voronikhin) Nowicka-Krawczyk, Mühlsteinová and Hauer* is a cyanobacterium of significant economic interest that is widely cultivated. The term “spirulina” is a common commercial name used for cyanobacteria belonging to the genus *Limnospira* or *Arthrospira*, which are grown for human consumption as food, nutraceuticals, and cosmetic products. However, this commercial name has led to misunderstandings with the scientific name of genus *Spirulina*, used for species not used as food [1]. Recently, the cultivated species of Spirulina were reclassified to the genus *Limnospira* by Nowicka-Krawczyk et al. [2], due to 16S rDNA genetic distance with respect to other species of *Arthrospira*.

This species has become a relevant food in human nutrition in recent times and has a tradition as a component of local dishes such as dihè in central Africa or tecuitalatl in Central America [1,2,3].

The presence of pigments specific to Cyanobacteria (and some eukaryotic algae like red algae and Glaucophyta), such as phycocyanin, has, as a consequence, a different capability of growth at given light wavelengths with respect to other cultivated algae.

The total amount of phycocyanin itself is influenced by light wavelength [4,5], as an adaptation to the spectrum at different depths in the water column (chromatic adaptation [6]).

In cultivation, the use of specific illumination instead of white light may change the growth speed and the nutritional content of cyanobacteria, due to direct influencing of the photosynthetic efficiency, and/or to the induction of primary and secondary metabolites accumulation. The use of various light wavelengths has been shown to trigger changes in pigment activity and activate genetic pathways leading to the biosynthesis of specific pigments, such as phycocyanin, which are better adapted to light capture [7].

It is hypothesized that different light wavelengths may also induce diverse storage activities in proteins, carbohydrates, lipids, and secondary metabolites, such as antioxidants. This hypothesis holds significant potential in the field of biotechnology, as the strategic use of different wavelengths for a specific duration could be employed to modulate Spirulina content for different nutritional purposes, resulting in the production of different functional foods tailored to different health effects.

The purpose of this article was to investigate the effects of different wavelengths of light on the growth of Spirulina, specifically blue light (460 nm) and yellow light (590 nm). The study aimed to determine whether these different types of illumination could cause changes in storage material in spirulina cells, with a focus on the concept of functional microorganisms [8]. The study also aimed to identify specific wavelengths that could induce changes in the chemical composition of Spirulina biomass, specifically by increasing the total amount of pigments or proteins. The most significant novelty of this work is its connection between biochemical data and ultrastructural observations, particularly with respect to changes in the number and structure of cytoplasmic inclusions.

## 2. Materials and Methods

### 2.1. Limnospira Fusiformis Growth and Experimental Set-Up

*Limnospira fusiformis* cells were grown in a climatized chamber AHSI (EKOCTL 900 P-ES, Angelantoni Ind. Spa, IT) under a white light at a medium intensity of PAR (Photosynthetically Active Radiation, 150 µE m/^2^ s/^1^) in a 2 L flask with Zarrouk medium [9] and bubbled for 10 min every hour by an aquarium pump. The nutritive level of the culture was maintained constant by appropriate dilution in order to avoid nutrient limitation. The dark/light period was 12 h:12 h, while the temperature was kept at 28 °C during the night, and at 29 °C during the day. In the flask, pH was in the range of 9.93–10.3 with a stable trend and salinity was between 10.7 and 14.6 psu.

To evaluate the response of *L. fusiformis* to different light qualities, the culture was divided into two subcultures and, successively, one was exposed to blue light (λ _max_ 460 nm) and one to yellow light (λ _max_ 590 nm), with PAR still being 150 μE m/^2^ s/^1^ as in Silva et al. (2020) and Sforza et al. (2012) [10,11]. These last authors showed that such intensity corresponded to the highest efficiency of light absorption by algal cultures. The energy flux is different at different wavelengths, but the number of photons is the factor to be considered, since a single photon contributes to each elementary photochemical act [12,13].

Then, 120 LEDs of 3000 and 6000 °K (SIRIO tape 11 Wm 860 24 V) were assembled in two panels. One was covered with a blue filter (max transmission 460 nm), and the culture vessel was surrounded by a highly reflecting diamond sheet to make the light field more homogeneous and reduce light dispersion. The distance between the vessel and the panels was adjusted to obtain the same intensity (150 µE/m^2^/s), and the distance between the vessel and the climatized chamber walls was 30 cm (Figure 1).

Then, the irradiance spectra (both of blue and yellow LED panels and of the white light used in the climatized chamber) were measured by an AvaSpec 2048 spectroradiometer and a CC3-UV-VIS cosine collector (Avantes B.V., Apeldoorn, NL) (Figure 2). The cultures were followed for 8 days, the first day being T0 (time 0), and after 8 days, T8. Morphological aspects were measured at the beginning (T0) and the end (T8) of the experiment, biochemical data were obtained at T0, T1, and T8, while in vivo absorption values were performed from T0 to T8.

### 2.2. Physical Measurements

#### 2.2.1. In Vivo Absorption and Growth Measurements

To carry out in vivo spectral absorption measurements and monitor cell growth during the experiment, we used an FDP-7UV200-2-5 immersion probe (Avantes, Apeldoorn, The Netherlands) that can measure spectral transmission through its own immersion in the fluid. The probe was connected to an AvaSpec 2048 spectrometer (Avantes), and the transmission (proxy for absorption in this highly absorbing medium) was directly measured in the suspension with a total optical path of 10 mm, as the space between the light emitting point and the reflecting surface was 5 mm.

The light source was AvaLight-HAL-mini (Avantes), and we acquired data using AvaSoft Basic Software.

Growth rates were calculated using absorption values at wavelengths of 440 nm, that is, the peak of chlorophyll and accessory pigments, 620 nm, that is, the peak of phycocyanin, and of 680 nm, the peak of chlorophyll *a*, using the following equation:$$\mathrm{Growth}\;\mathrm{rate}=\frac{ln\frac{x}{x_{0}}}{ln2}\times \frac{24}{\mathrm{Actual}\;\mathrm{hours}}$$
where:X = the absorption measure of one day;X_0_ = the absorption measure of the previous day.

Also, a linear model of the growth rate was used, but no difference was found.

#### 2.2.2. Chemicophysical Conditions

Through the multiparametric HL4 probe (Hydrolab srl, Matera, Italy), which brings together several sensors and electrodes, daily measurements of dissolved oxygen, salinity, pH, and ORP (oxidation reduction potential) on *L. fusiformis* were recorded.

### 2.3. Biochemical Analysis

#### 2.3.1. Extraction and Analysis of Liposoluble Pigments

Chlorophyll *a* (Chl) extraction was performed according to Lazzara et al. (2010) [14]. A known amount of culture (usually 20 mL) was filtered on a Whatmann GF/F 25 mm glass fiber filter, and the acetone (90%) extract was measured after one day with a SHIMADZU UV-2600 spectrophotometer. The absorption spectrum was measured between 350 and 750 nm.

The concentrations of chlorophyll and carotenoids were calculated according to the formulas of Jeffrey and Humphrey (1975) [15], Richards and Thompson (1952) [16], and Strickland and Parsons (1968) [17].

#### 2.3.2. Phycocyanin Extraction and Analysis

The method for phycocyanin (PC) extraction is similar to chlorophyll extraction. The difference is that the homogenization is made with 7 mL of phosphate buffer (pH 7). A known amount of culture (usually 20 mL in three replicates) was filtered on a Whatmann GF/F 25 mm glass fiber filter, and the extract was measured with a SHIMADZU UV-2600 spectrophotometer. The absorption spectrum of the extract was measured between 350 and 750 nm.

The calculation of the PC concentration was performed according to Kursar and Alberte (1983) [18], using the absorption at 618 nm.

#### 2.3.3. Protein Extraction and Analysis

Protein extraction was carried out using the Bradford method (1976) [19]. This method involves the reaction of the chromogen Coomassie Brilliant Blue G-250, which binds to the proteins in the sample. A known amount of culture (usually 20 mL in three replicates) was filtered on a Whatmann GF/F 25 mm glass fiber filter. Then, the filter was homogenized with 2.5 mL of NaOH 0.5 N, and it was left in a thermostatic bath at 60 °C in the oven. After two hours, the sample was centrifuged at 4000 rpm for 20 min. Finally, 100 μL of protein extract was added to 5 mL of chromogen Coomassie Brilliant Blue G-250 (PanReac AppliChem, Darmstadt, Germany) to make the reaction develop. The absorbance was measured at a wavelength of 595 nm with a SHIMADZU UV-2600 spectrophotometer (Shimadzu, Kyoto, Japan).

#### 2.3.4. Determination of Suspended Solids and Ashes

The Van der Linde method (1998) [20] was used to determine the suspended solids (dry weight) and ashes in a given volume (20 mL) of Spirulina culture.

Three weights were obtained for each filter—white, dry weight, and calcined weight—and were all used to calculate the percentage of suspended solids and organic matter of the analyzed sample.

Before carrying out the filtration, we weighed GF/F glass fiber filters with a diameter of 25 mm to obtain the weight of the “white” filters only. We then filtered 20 mL of the culture for the three replicates of each sample. The filters were placed in the oven for 24 h at 60 °C [21], then in a vacuum desiccator at room temperature for another 24 h. Lastly, we weighed the filters again. Three consecutive weightings produced the same result, showing that the water had completely evaporated. All weights were measured on a Sartorius electronic scale. To determine the calcined weights and the organic matter, once we weighed the filters, we placed them in a muffle for 24 h at a temperature of 450 °C. Then, we kept them in a desiccator for another 24 h. This allowed us to determine the weight of the calcined filters.

#### 2.3.5. Statistical Analysis

For all biochemical and growth rate measurements, statistical comparisons were carried out using the T-student method. The difference was considered significant for *p* < 0.01.

### 2.4. Microscopic Analysis

#### 2.4.1. Microscope Observation and Filaments Count

Next, 10 mL of culture, fixed with 3 mL of formalin 1%, were collected for the observation under the microscope. We used a microscope Zeiss IM35 (Carl Zeiss, Oberkochen, Germany) with a hemocytometer to count the *L. fusiformis* filaments, measure the cell body length, and count the number of turns.

#### 2.4.2. Fixation and Inclusion by Electron Microscope

The fixation and inclusion in the Spurr resin [22], for observation under the electron microscope, were performed following the methods described by Kimura et al. (2012) [23] and Yau et al. (2016) [24] as adapted to cyanobacteria, as shown in Capelli et al. (2017) [25] and Papini et al. (2017, 2018) [26,27]. Culture samples were collected with the growth medium and fixed in 1.25% glutaraldehyde in filtered sea water at 4 °C, then post-fixed in 1% OsO4 in 0.1 M phosphate buffer at pH 6.8. After dehydration in an ethanol series and a further step in propylene oxide, the samples were embedded in Spurr’s epoxy resin. At each step, the cells were centrifuged for 5 min at 1500 rpm. Only the sediment was used for the following step in order to substitute the solvents without collecting too many filaments with the pipettes.

Cross sections about 80 nm thick were cut with a diamond knife (with a Reichert-Jung Ultracut ultramicrotome). The sections were stained with uranyl acetate and lead citrate, and then observed with a Philips EM201 TEM (Philips, Worcester, MA, USA) at 80 kV. Living samples and semi-thin sections were observed with a light microscope.

## 3. Results

### 3.1. Biochemical Data

At days T0, T1, and T8, biochemical analysis and measurement of dry weight were carried out. The concentrations of proteins, dry weight, chlorophyll *a*, carotenoids, and PC are shown in Table 1, where the average values and the standard deviation are reported. As we can see in Figure 2a,b,e, yellow light stimulated the production of proteins, total biomass, and PC with percentages that, at T8, were, respectively, +104%, +77%, and +38% compared to T0. If instead, we compare the data at T8 (Table 1) between yellow and blue light, the ratio for protein and dry weight were, respectively, 190% and 187%. For what concerns PC (Table 1 and Figure 3e), the concentration in blue light increased immediately at T1 (12.16 µg/mL), but it decreased at T8 to the value of 5.86 µg/mL, while in yellow the increase was delayed to T8, when it was almost twice as much that in blue light.

Chl concentrations were similar in yellow and blue light at T1, with a slight increase (20%) in yellow at T8, and a decrease in blue (Figure 3c). The carotenoids concentration (Figure 3d) was unchanged in blue, while in yellow it increased by 52% (0.99 µg/mL T1—1.51 µg/mL) at T8.

### 3.2. Filaments Counts

Filaments counts were made using the optical inverted microscope with a hemocytometer. One sample at T0 and two samples at T8, one of the culture grown under blue light and one grown under yellow light, were examined. From the observations, the total number of filaments/L in blue light was similar to the mother culture (T0), while in yellow light a higher number was observed, about 54% more than in blue. Moreover, the yellow culture had a higher percentage of spiraled filaments, which were also longer than the blue light-grown ones.

### 3.3. Growth

Figure 4 shows the daily growth rates, which were obtained from the average in vivo absorption at 440, 620, and 680 nm, as a function of time.

Even though the growth rates were similar in the two cultures at T1, they changed significantly in the following days. At T2 and T3, the yellow light stimulated the production of biomass (Figure 4). From T6 to T8, the growth slowed down, also because the culture medium had not been changed in the previous day.

### 3.4. Ultrastructure

*L. fusiformis* exhibited a filamentous arrangement, consisting of a variable number of cells forming a helical structure.

Under white light (control), trichomes had an average width of 6 μm, with varying heights ranging from 3 to 6 μm, also due to the formation in some points of cell walls during cell divisions. Some polyhedral bodies (from here on carboxysomes) were observed in the cytoplasm.

When exposed to yellow light, the cells of the filament had an average width of 5.7 μm and an average height of 2.6 μm relative to the main axis of the filament (Figure 5A). The cells height ranged from 2 μm in the middle to about 3 μm along the longitudinal tangential walls, which was not significantly different from time T0. Grey (medium electron density) ovoidal bodies measuring approximately 5–600 nm (Figure 5A,B) were on average more numerous than in the control. In general, the grey bodies (from here on cyanophycin granules) tended to be positioned near the dividing wall that separates two cells of the filament. Notably, expanded thylakoids were observed in contact with the cyanophycin granules (Figure 5B), and the thylakoids were more abundant near the external walls, often exhibiting dilations resembling vesicular structures (Figure 5C). In the final two cells of the filaments, the thylakoids were disposed in longitudinal position, starting from the apex of the filament (Figure 5D). Some cyanophycin granules appeared to be surrounded by a membrane (Figure 5D and Figure 6A). Electron transparent bodies appeared between bundles of thylakoids (Figure 6A). Here, the cytoplasm was dense with ribosomes, while granules larger than ribosomes also appeared, particularly close to the dividing wall between two cells of the filament.

Towards the end of the filament, carboxysomes were found to be more commonly distributed, occupying a position approximately in the middle of the cell cytoplasm (as depicted in Figure 6A). Additionally, circular structures were also observed.

Interestingly, in the presence of blue light, the previously abundant grey bodies (which were prominent in yellow light) significantly decreased in number, whereas the electron-dense bodies (carboxysomes) appeared to be more abundant (Figure 6B). When observed at higher magnification, we observed dilated thylakoids coming into close proximity with the dividing walls (Figure 6C). Furthermore, smaller grey bodies (cyanophycin granules) measuring approximately 3–400 nm were observed in the cytoplasm (Figure 6C). Notably, the thylakoids were arranged in a radial pattern emanating from the dividing wall (Figure 6D), and did not show an evident enlargement of the lumen, which was observed in yellow light.

## 4. Discussion

As observed in the results, notable differences were observed at the end of the experiment between the effects of blue and yellow light. Yellow light stimulation resulted in higher biomass production compared to blue light, as depicted in Figure 3 in terms of growth rates, and as for filaments (in Section 3.3) where their total amount at T8 is higher than in blue light. In addition to this, the filaments cultured in yellow light exhibited a higher percentage of spiraled trichomes, which were also longer compared to those cultured in blue light. It is worth mentioning that morphological modifications in Spirulina are often observed in response to changes in environmental factors such as temperature and medium composition [28,29,30,31]. Furthermore, the linearization of Spirulina filaments frequently occurs in laboratory and mass cultures, particularly in depleted medium [29], and is considered as one of the physiological morphologies of *L. fusiformis* [3,32,33].

The growth rates also confirmed the development of a higher biomass in yellow light (Figure 4); in fact, also at T2 and T3, the yellow light induced more than one div/day, nearly double that in blue light.

The increase in biomass with yellow light has also been observed by Markou (2014) [34] and Walter (2011) [35], who studied the growth of *L. fusiformis* biomass cultivated with Zarrouk medium and illuminated with yellow, blue, and red and green lights in low PAR.

Kumar at al. (2021) [36] cultivated *L. fusiformis* in different spectral light quality and observed an increase with orange light in dry weight (61%), Chl (40%) and carotenoids (29%), with respect to white light control. We have observed with yellow light an increase of dry weight (76%), Chl (19%), and carotenoids (89%) with respect to T0 (Figure 4).

In addition to the production of biomass, yellow light also stimulates the production of proteins (Figure 3). Considering proteins in relation to biomass, the percentage increased from 34% at T0 to 40% at T1 in yellow light, and this was maintained at T8 (Figure 7). On the other hand, in blue, after a slower rise—just 1% more at T1 compared to T0 (34%)—the protein concentration becomes similar to that in the yellow light (39.3%) at T8. The protein/dry weight ratios (Figure 7a) are comparable with data indicated in literature, which range between 40 and 65%, being close to the lower limit [31,37,38].

In general, the culture medium also influences the growth of *L. fusiformis* [39,40]. For example, higher ratios between proteins and dry weight are obtained by using urea as a source of nitrogen [41]. In addition to the medium, the growth environment of the culture can also influence the protein content. An example is the fact that, in laboratory experiments, protein concentration appears to be lower (37%) than in outdoor tank growth (58%) [42]. This could justify the not-too-high yield of the protein content found in our experiment.

The relative concentration of chlorophyll *a* with respect to both DW and protein is slightly higher in blue than in yellow light (Figure 7b,d), although the reverse occurs when taking into account the absolute concentrations at T8. In blue light, the Chl/DW ratio remains more or less the same as in T0, while in yellow light a decrease occurs, already at T1, which is confirmed at the end of the experiment, when the lowest ratio is recorded (0.83%). The concentrations found are comparable with those of the results by Soni (2012) [38], Olaizola (1990) [43], and Ciferri (1983) [3], while Chen et al. (2010) [44] found the best increase of chlorophyll just in yellow light. These last authors, however, used much stronger light intensities (750 or 1500 μmol m^−2^ s^−1^) in their experiment.

The increase in carotenoids observed in yellow light could be explained by the fact that chlorophyll cannot capture light energy directly from the light source, but receives it from carotenoids and phycobilins as excitation energy [45].

For what concerns phycocyanin (PC), we can see that the ratios with DW and proteins at T1 are higher in blue light (Figure 7c,e).

This result was unexpected since generally phycocyanin synthesis should be stimulated by yellow light (red light, more specifically) as a result of complementary chromatic adaptation [46,47] and not by blue light. Nevertheless, phycobilins synthesis can also occur as a result of non-complementary chromatic adaptation or inverse chromatic adaptation [48,49,50]. The explanation of this phenomenon is not straightforward, but Kirk (1994) [10] proposed that it might be induced not by the specific spectral nature of the employed light, but rather by the imbalance in excitation energy received by the two photosystems. In any case, non-complementary adaptation in blue light may stimulate phycobilisome synthesis only at the beginning, while complementary adaptation with a decrease in phycocyanin synthesis would occur later (as observed at T8 in our experiment).

At T8, the PC/DW and PC/Prot ratios are similar in yellow and blue light, as expected by literature [51]. Also, in the PC/Chl ratio, the highest value at T1 is in blue light (Figure 7f). Our results are not unique, since an increase in phycocyanin in Spirulina exposed to blue light was observed also by Chen et al. [41], who did not comment on this result.

The ultrastructural data showed an increase in grey bodies in yellow light with respect to the control and blue light. These bodies could be identified as cyanophycin granules on the basis of the morphology described by Van Eykelenburg (1979) [26]. After this author, cyanophycin granules tend to be more abundant at low temperature (15–17 °C) with low light intensities in *L. fusiformis.* Cyanophycin granules have been interpreted as nitrogen-storing bodies (Simon 1973) [52], and after Van Eykelenburg (1979) [26], the storage in cyanophycin granules may be linked to a reduction of protein synthesis at low temperature/low light intensity, while their number decreases in a state of optimal growth (higher than 17 °C) and even in the case of exposition to blue light, as in our experiment. Cyanophycin granules, after Flores et al. (2019) [53], are formed by poly-aspartate-arginine material and are stored by cyanobacteria in the case of unbalanced growth but with sufficient availability of nitrogen. The formation of cyanophycin granules is not linked to ribosomal activity [54] and, as a matter of fact, in our observations the density of ribosomes in yellow light appeared lower than in blue light. These observations may suggest that the increase in protein content observed by several authors in blue light may be related to the increase in C-phycocyanin [54], while the higher amount of proteins in yellow light during the first days of our experiment are to be attributed to cyanophycin granules.

The enlargement of the intrathylakoidal space observed in yellow light has been observed in the case of stress, for instance, by cadmium [55]. The electron transparent bodies were interpreted as polyhydroxybutyrate grains by comparison with previous observations in some species of genus *Cyanothece* [56]. Since at least some cyanophycin granules appeared surrounded by a unit membrane, they were considered as inclusions in largely dilated intrathylakoidal spaces [56]. In spirulina, however, a unit membrane was observed only for some of the granules (see, for instance, Figure 5D), and hence some of these granules may be free in the cytoplasm.

The contemporaneous minor amount in carboxysomes observed in yellow light with respect to blue light may be linked to a reduction in Calvin Cycle activity, since the function of carboxysomes is to increase the concentration of CO_2_ around Rubisco [57,58].

In blue light, the number of cyanophycin granules is lower, while there are more carboxysomes. The identification of carboxysomes and carboxysome in formation (circular structure in Figure 5D) was done on the basis of the images in Orùs et al. (1995) [59].

## 5. Conclusions

Finally, we can say that yellow light has an effect on increasing the production of dry weight and proteins, and apparently, most of the increase in proteins is to be attributed to the higher amount of cyanophycin granules.

Our results highlight the biotechnological potential of this approach: in fact, using specific wavelengths in the harvesting phase (for one or a few days) of *L. fusiformis* growth, it is possible to increase the content of phycocyanin after one day of exposure to blue light, that is one generation time, and to optimize the harvesting of proteins and dry weight with yellow light, after 5 to 6 generation times.

## Figures and Tables

**Figure 1 microorganisms-11-01236-f001:**
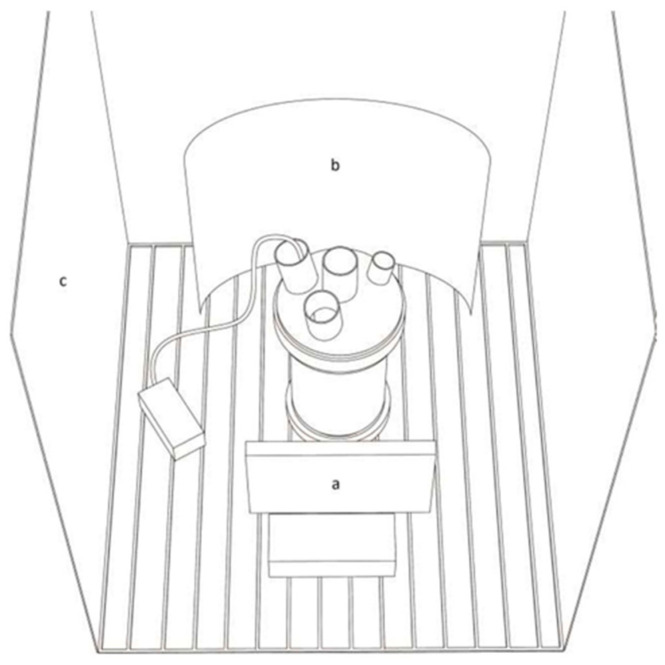
Schematic representation of a culture vessel in the climatized chamber. Light source (a), reflection surface (b), chamber white walls (c).

**Figure 2 microorganisms-11-01236-f002:**
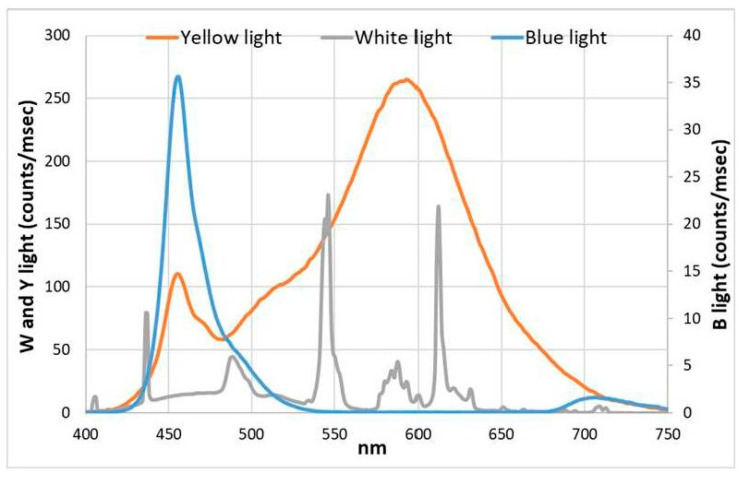
Irradiance spectra of two LED light sources and of the climatized chamber white light.

**Figure 3 microorganisms-11-01236-f003:**
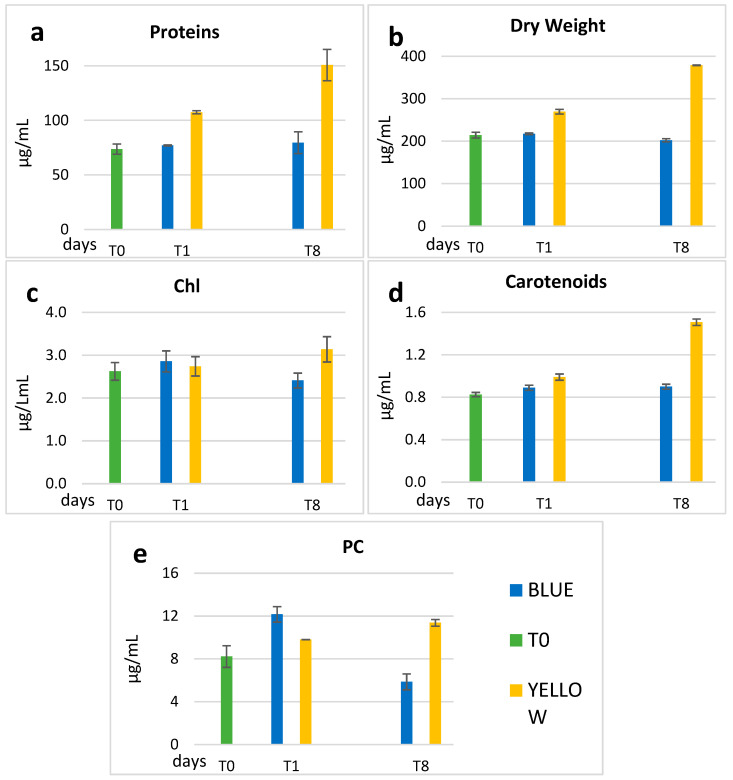
Concentrations of proteins (**a**), Dry weight (**b**) and pigments (**c**) Chl, (**d**) Carotenoids, (**e**) PC, at T0 (green bar) with climate chamber’s white light, and in the blue (blue bar) and yellow (yellow bar) light experiment.

**Figure 4 microorganisms-11-01236-f004:**
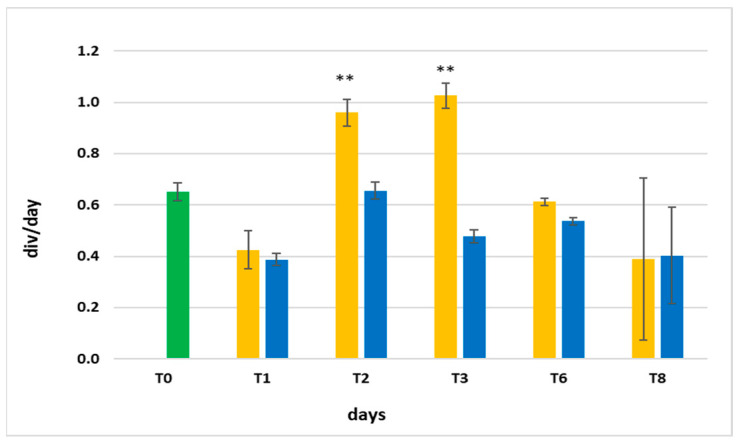
Daily growth rates obtained from the average in vivo absorption at 440, 620, and 680 nm starting from T0 (green bar) during the experiment with blue light (blue bar) and yellow light (yellow bar). ** Significant difference (*p* < 0.01) between yellow and blue data at T2 and T3.

**Figure 5 microorganisms-11-01236-f005:**
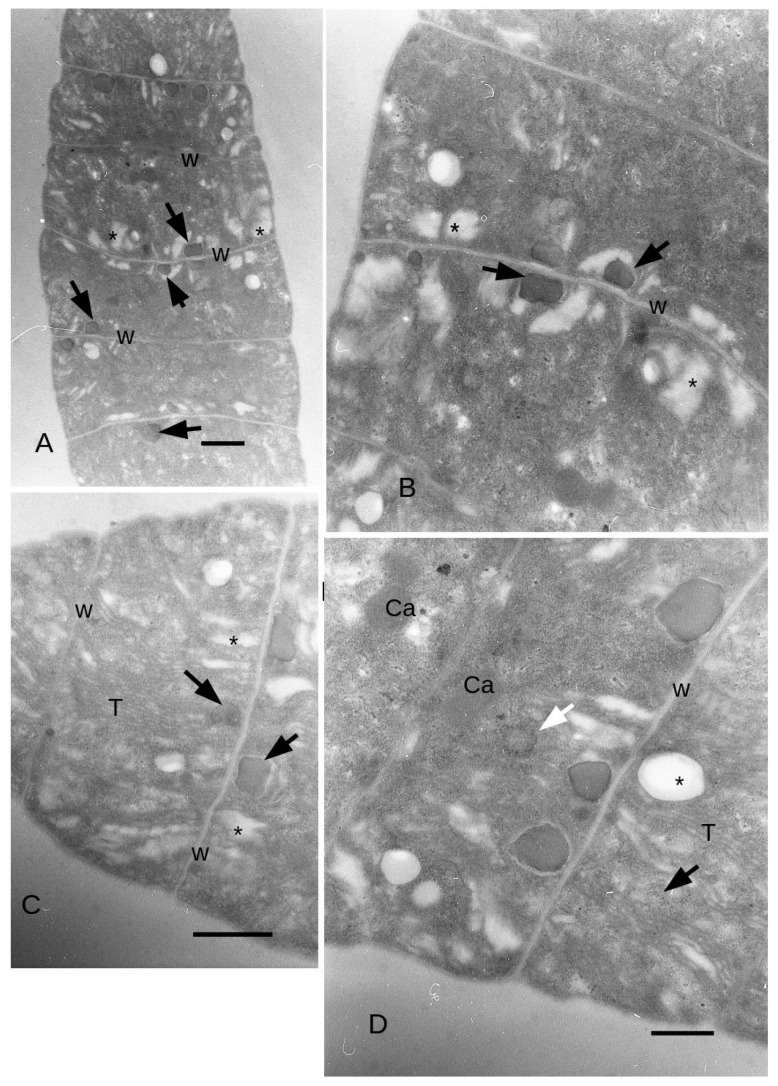
*L. fusiformis* filaments grown in yellow light (8 days). TEM images. (**A**): The filament width is 5.7 μm on average, while the cells’ height ranges from 2 μm in the middle of each cell to about 3 μm along the longitudinal tangential walls. Cyanophycin granules (arrowheads) are common in the cytoplasm. In many points close to the walls, some thylakoids are expanded (asterisks). Bar = 1 μm. (**B**): Cyanophycin granules (arrows) are common close to the transversal walls, adjoining thylakoids expansions. Bar = 1 μm. (**C**): In the final cells of the filaments, thylakoids membrane extends in the middle of the cell. Bar = 1 μm. (**D**): Some electron transparent bodies (asterisk) are formed between bundles of thylakoids surrounded by many ribosomes (arrows). Bar = 500 nm. Ca = Carboxysomes; Cy = Cyanophycin granules; T = Thylakoids; W = Wall.

**Figure 6 microorganisms-11-01236-f006:**
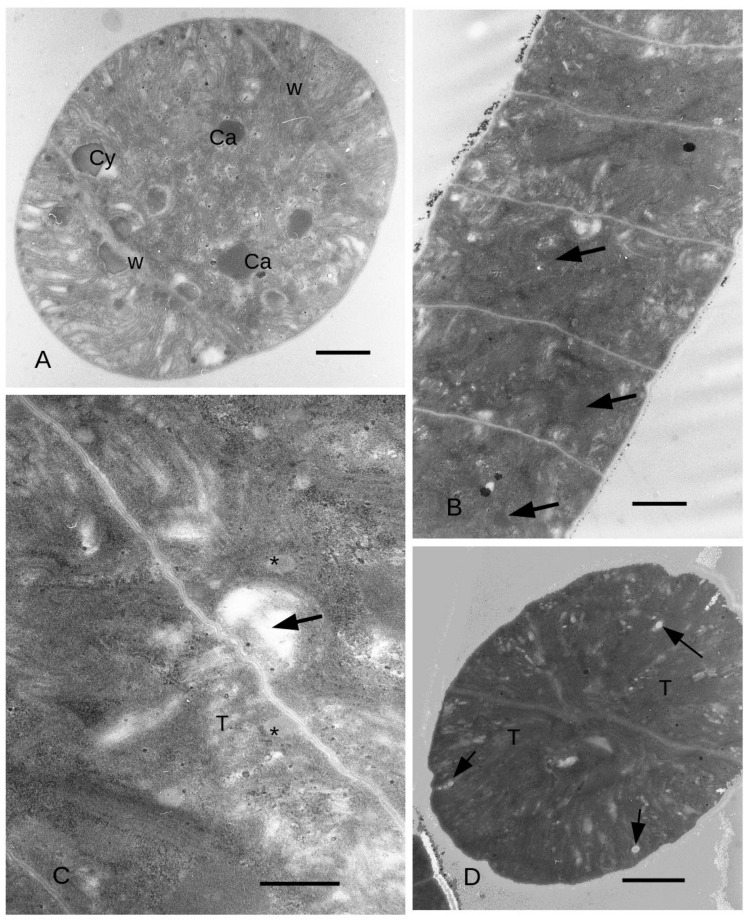
*L. fusiformis* filaments grown in yellow light and blue light. TEM images. (**A**): filament grown in yellow light. Close to the end of the filament, carboxysomes are more common. Circular structures are present, interpreted as carboxysome in formation. Bar = 1 μm. (**B**): Filament grown in blue light. Electron-dense bodies (arrows) are observed in the cytoplasm. Bar = 1 μm. (**C**): Filament grown in blue light. Some thylakoids (asterisks) are very close to the transversal walls, and at one point an electron transparent material (arrow) is forming. Bar = 500 nm. (**D**): Filament grown in blue light. Close to the end of the filament, the thylakoids are in general radially oriented, starting from the dividing wall. White electron transparent granules (arrows) are forming. Bar = 1 μm.

**Figure 7 microorganisms-11-01236-f007:**
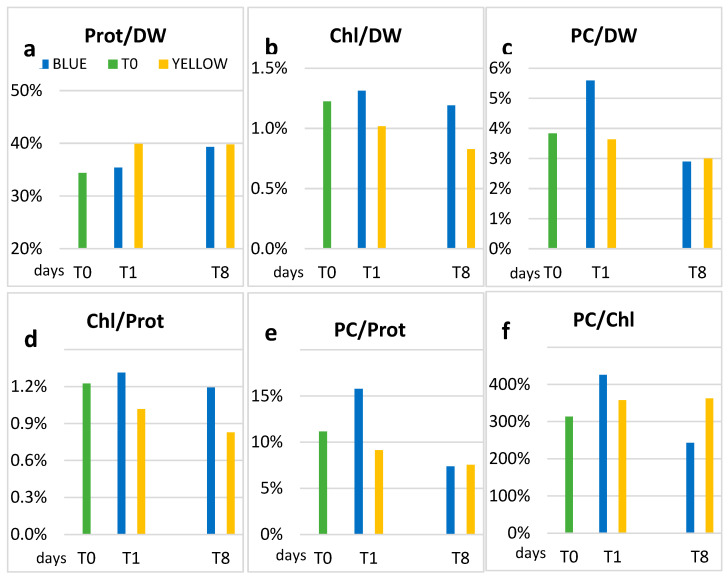
Protein and Dry Weight (**a**), Chl and Dry Weight (**b**), PC and Dry Weight (**c**), Chl and Protein (**d**), PC and Protein (**e**), PC and Chl (**f**) ratios obtained from the data of the absolute concentrations at T0 (green bar) with climate chamber’s white light, and in the blue (blue bar) compared to yellow (yellow bar) light experiment.

**Table 1 microorganisms-11-01236-t001:** Pigments, proteins concentrations, and dry weight (DW) at T0 (in white light climatized chamber) and in blue and yellow light experiment. ** significant difference (*p* < 0.01) between yellow and blue data at T8.

		Chl	CAR	PC	SD	Proteins	SD	DW	SD
		µg/mL	µg/mL	µg/mL		µg/mL		µg/mL	
	T0	2.62	0.83	8.22	1.01	73.69	4.31	214.42	6.68
Blue	T1	2.86	0.89	12.16	0.72	77.06	0.50	217.60	2.07
	T8	2.41 **	0.90	5.86 **	0.74	79.52 **	9.30	202.3 **	3.69
Yellow	T1	2.74	0.99	9.80	0.00	107.46	1.32	269.51	5.64
	T8	3.14 **	1.51	11.37 **	0.31	150.7 **	13.36	378.9 **	0.94

## Data Availability

Not applicable.

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
