# Peer review of "Blue and Yellow Light Induce Changes in Biochemical Composition and Ultrastructure of Limnospira fusiformis (Cyanoprokaryota)"

_microorganisms, 2023, doi:10.3390/microorganisms11051236_

Round 1

Reviewer 1 Report

introduction

in line 42. Please also include red algae, since they also produce PBPs

in lines 45-46. "influence-influencing" is redundant. please use synonyms

please add a paragraph were explain the importance of the application of sécific wavelenght on the production of PBPs

Materials and methods

please specify how many samples were taken and how oftn all the parameters were measured

Results

figure 3 and 4 must be improved. is it the green bar the control?

for cell count comparison a simple ANOVA would suffice to determine if there is statistical difference between them. Please add it

why the growth was measured until day 6, but the metabolites were measured only in day 1 and day 8?

figure 4 mention statistical significance, how was it determined?. please add the statistical method used

figures 5 and 6 mentions A platensis, while the work is with L fusiformis, which one is correct?

table 3 and figure 7 show the same data, choose only one way to present the data

Author Response

Introduction
in line 42. Please also include red algae, since they also produce PBPs
We include red algae as suggested by the reviewer (Line 42-43).
in lines 45-46. "influence-influencing" is redundant. please use synonyms
We have introduced a synonym of “influence”.
please add a paragraph were explain the importance of the application of sécific wavelenght on the
production of PBPs
We added a specific paragraph with two new literature entries:
Fujimori E, Pecci J (1966) Dissociation and association of phycocyanin. Biochemistry 5(1): 3500-3508.
Hsieh-Lo, M., Castillo, G., Ochoa-Becerra, M. A., & Mojica, L. (2019). Phycocyanin and phycoerythrin:
Strategies to improve production yield and chemical stability. Algal Research, 42, 101600.
Materials and methods
please specify how many samples were taken and how oftn all the parameters were measured
We added the requested information in the paragraph 2.1 from line 110 to line 112.
Results
figure 3 and 4 must be improved. is it the green bar the control?
We improve the explanation of the captions of both Figure 3 and 4.
for cell count comparison a simple ANOVA would suffice to determine if there is statistical difference
between them. Please add it
For our purpose, the qualitative analysis was sufficient. For this reason, we included the data description in
the text from line X to X.
why the growth was measured until day 6, but the metabolites were measured only in day 1 and day 8?
Following the suggestion of the reviewer, we revised the spectral absorption data to calculate the growth
rate at T8 including it in Figure 4 and the text.
figure 4 mention statistical significance, how was it determined?. please add the statistical method used
We have added a new paragraph regarding the statistical analysis.
figures 5 and 6 mentions A platensis, while the work is with L fusiformis, which one is correct?
We corrected this point: the first one is the old name.
table 3 and figure 7 show the same data, choose only one way to present the data
Figure 7 was chosen to present the data and table 3 was cancelled.

Reviewer 2 Report

1-   The topic is timely and worth revision. However, the paper lacks a coherent structure and in its present form is merely a compilation of information without liason and critical analyses.

2-   The title is too long and should be shortened and concentrated.

3-   The authors claim that the protein content increased under yellow light and it is documented for tens of years that blue light increases the protein content in algae and cyanobacteria. The authors should explain.

4-   Some expressions in the whole article should be changed. For e.g. the authors used thalli for the count of the filaments. Also in the authors mentioned 2.4.1. Microscope observation and cell count. The spirulina is a filament and not a cell. Also, how did you count the number? What instrument did you use other than the microscope for e.g., hemacytometer, or any other measuring tools?

5-   L 169, 10 ml of colture. What colture???

6-   Figure 5 and 6,  the TEM images are not good quality enough to represent differences

7-   Most of the figures are in low quality

8-   The English language over the whole manuscript is not of a good standard and should be extensively edited by a native English speaker.

9-   Also, scientific English quality requires revision

10-    The novelty of the work is not addressed well by the authors. The subject is not new and lots of papers published on the effect of light quality and types on algae and cyanobacteria  

Author Response

Reviewer 2
1-The topic is timely and worth revision. However, the paper lacks a coherent structure and in its present form is merely a compilation of information without liason and critical analyses.

We improved the critical analysis in the discussion adding also 5 new literature entries.

2-The title is too long and should be shortened and concentrated.

We shortened the title as suggested by the reviewer.

3-The authors claim that the protein content increased under yellow light and it is documented for tens of years that blue light increases the protein content in algae and cyanobacteria. The authors should explain.

We added a paragraph (on pg 11) about this point with a new reference entry.

4-Some expressions in the whole article should be changed. For e.g. the authors used thalli for the count of the filaments. Also in the authors mentioned 2.4.1. Microscope observation and cell count. The spirulina is a filament and not a cell. Also, how did you count the number? What instrument did you use other than the microscope for e.g., hemacytometer, or any other measuring tools?

We agree with the reviewer, thus we changed from “thallus” (this term was used in some old reference) to “filament” and we made clear when we intend to discuss the filament size or the filament’s cell size. We used microscope with a hemocytometer and this methodology was specified in the paragraph: Materials
and methods.

5-L 169, 10 ml of colture. What colture???

We thanks the reviewer for his/her suggestion and we corrected it.

6-Figure 5 and 6, the TEM images are not good quality enough to represent differences

We check it: probably it depends on pdf compression.

7-Most of the figures are in low quality
We check it: probably it depends on pdf compression.

8-The English language over the whole manuscript is not of a good standard and should be extensively
edited by a native English speaker.

We recheck extensively the English language.

9-Also, scientific English quality requires revision

We recheck extensively also the scientific English.

10-The novelty of the work is not addressed well by the authors. The subject is not new and lots of papers
published on the effect of light quality and types on algae and cyanobacteria

The strength of the work is the connection between biochemical data and ultrastructural observations that can provide a clarification about the cellular changes with particular reference to the cytoplasmic inclusions: we cleared this point at the end of the introduction. 

Reviewer 3 Report

This is a straightforward descriptive paper, presenting some thorough analysis of the growth, composition and morphology of Limnospira fusiformis grown under different light colours. It doesn't give any deep insight, but it is fine as far as it goes. One very minor correction: line 67-68 - PAR normally stands for Photosynthetically Active Radiation.

Author Response

Reviewer 3
This is a straightforward descriptive paper, presenting some thorough analysis of the growth, composition and morphology of Limnospira fusiformis grown under different light colours. It doesn't give any deep insight, but it is fine as far as it goes. One very minor correction: line 67-68 - PAR normally stands for
Photosynthetically Active Radiation.

Corrected: thank You! We improved the critical analysis in the discussion adding also 5 new literature entries. 

Reviewer 4 Report

On manuscript on Pelagatti et al. on improvement on light conditions on culture Limnospira fusiformis (spirulina) on biotechnological applications. On start what on needs on biotechnological applications demands on L. fusiformis. There on a description on the introduction section on comprehending the findings on study. Another remark on why on blue and yellow light chosen and why not on other light intensities despite previous findings? Another aspect refers on the culture medium conditions. Could these on contributed on the results on study? Why were not confronted the data on yellow and blue light on white light? Why on light a need on improvements? Could other variables permit on obtaining these results or even better on culture improvements? Other aspect refers on the costs on applying the blue and yellow light on comparison on white light?

Finally, on biotechnological applications time on a limiting parameter?

Author Response

Reviewer 4
On manuscript on Pelagatti et al. on improvement on light conditions on culture Limnospira fusiformis (spirulina) on biotechnological applications. On start what on needs on biotechnological applications demands on L. fusiformis. There on a description on the introduction section on comprehending the findings on study.

We improved the introduction explaining better the aim of the study and the importance of the results.

Another remark on why on blue and yellow light chosen and why not on other light intensities despite previous findings?

We linked to previous studies about blue and yellow light: from a biotechnological point of view, blue and yellow LED light are among those easier to obtain at low cost.

Another aspect refers on the culture medium conditions. Could these on contributed on the results on study?

The reviewer is surely right: the culture medium conditions are able to influence largely the results and we will plan a future experiment about that. In this more limited investigation, we kept the culture conditions as constant as possible using a very standard medium as the Zarrouk medium. We explained it better in the text.

Why were not confronted the data on yellow and blue light on white light? Why on light a need on improvements?

We compared the data also to white light: however, white light is the “normal” condition and results about white light growth in Zarrouk medium are very abundant. We explained it better in the text.

Could other variables permit on obtaining these results or even better on culture improvements? Other aspect refers on the costs on applying the blue and yellow light on comparison on white light?

As written above, we agree with the reviewer that other parameters would surely change the results. We tried therefore to keep as constant as possible the medium. We explained it better in the text.

Finally, on biotechnological applications time on a limiting parameter?

We improved the conclusion with this suggestion by the reviewer: the idea is that a final step of few days (or only one day) with a specific light may be used to change the nutritional content of spirulina in the direction of the increase in phycocyanin or proteins or other pigments. 

Round 2

Reviewer 1 Report

The authors have significantly improved the quality of the document

Reviewer 2 Report

I have gone through the revised version of the manuscript and I found that the authors revised their Ms, and it may be acceptable after English corrections

Reviewer 4 Report

Authors on improved manuscript and now warrants publication on Microorganisms.